



# Low-level mixed-phase clouds at the high Arctic site of Ny-Ålesund: A comprehensive long-term dataset of remote sensing observations

Giovanni Chellini[1], Rosa Gierens[1], Kerstin Ebell[1], Theresa Kiszler[1], Pavel Krobot[1], Alexander Myagkov[2], Vera Schemann[1], and Stefan Kneifel[1, 3]

[1]Institute for Geophysics and Meteorology, University of Cologne, Cologne, Germany
[2]Radiometer Physics GmbH, Meckenheim, Germany
[3]Meteorological Institute, Ludwig-Maximilians-University Munich, Munich, Germany

**Correspondence:** Giovanni Chellini (g.chellini@uni-koeln.de)

**Abstract.** We present a comprehensive quality-controlled 15-month dataset of remote sensing observations of low-level mixed-phase clouds (LLMPCs) taken at the high Arctic site of Ny-Ålesund, Svalbard, Norway. LLMPCs occur frequently in the Arctic region, and extensively affect the energy budget. However, our understanding of the ice microphysical processes taking place in these clouds is incomplete. The dual-wavelength and polarimetric Doppler cloud radar observations, which are the cornerstones of the dataset, provide valuable fingerprints of ice microphysical processes, and the high number of cases included allows for the compiling of robust statistics for process studies. The radar data are complemented with thermodynamic retrievals from a microwave radiometer, liquid base height from a ceilometer and wind fields from large-eddy simulations. All data are quality controlled, especially the cloud radar data, which are accurately calibrated, matched, and corrected for gas and liquid-hydrometeor attenuation, ground clutter and range folding. We finally present an analysis of the temperature dependence of Doppler, dual-wavelength, and polarimetric radar variables, to illustrate how the dataset can be used for cloud microphysical studies. The dataset has been published in Chellini et al. (2023) and is freely available at: www.doi.org/10.5281/zenodo.7803064.

## 1 Introduction

Clouds are an essential component of the Earth system, considerably impacting the energy budget and driving the hydrological cycle (Wallace and Hobbs, 2006). As such, they are thought to play a role in the enhanced warming observed in the Arctic region in the past decades, termed Arctic Amplification (Shupe and Intrieri, 2004; Serreze and Barry, 2011; Matus and L'Ecuyer, 2017; Tan and Storelvmo, 2019). Clouds in the Arctic display features unique to this region, in particular the widespread and frequent occurrence of low-level mixed-phase clouds (LLMPCs) (Morrison et al., 2012; Mioche et al., 2015; de Boer et al., 2009; Shupe, 2011). Arctic LLMPCs are typically characterized by a shallow liquid layer at cloud top, where ice crystals form and grow into precipitation (Shupe et al., 2006; Morrison et al., 2012; Chellini et al., 2022). The liquid layer sustains itself from the continuous mass loss due to precipitation via turbulence-driven condensation; turbulence and updrafts being in turn produced buoyantly by cloud-top radiative cooling (Solomon et al., 2011; Morrison et al., 2012).



A variety of questions on the macro- and microphysical processes determining the radiative and thermodynamic characteristics, as well as the organization, phase-partitioning, and precipitation intensity of Arctic LLMPCs remains unanswered (Shupe et al., 2022; Wendisch et al., 2023) In particular, ice microphysical processes and their interaction with the liquid phase and turbulence are expected to affect precipitation, and therefore to determine the mass sink of the cloud layer (Morrison et al., 2012; Solomon et al., 2014, 2015). Harrington and Olsson (2001) and Simpfendoerfer et al. (2019) have in fact suggested lower cloud fractions and faster dissipation for Arctic mixed-phase stratocumuli that develop precipitation. It is speculated that precipitation-induced cold pools at the surface lead to thinning and break-up of the organization in Arctic stratocumuli (Abel et al., 2017; Eirund et al., 2019). Moreover, model sensitivity experiments have shown that the phase partitioning of Arctic LLMPCs is strongly sensitive to the assumptions on the habits of the ice particles, via mass-size, and size-fall speed relations (Avramov and Harrington, 2010). Sotiropoulou et al. (2022) recently showed that a more realistic representation of secondary ice production and ice aggregation processes in the NorESM2 climate model leads to an improvement in the degree of agreement with observations, especially ice water content (IWC) retrievals. Furthermore, the magnitude of Arctic Amplification itself in the Community Earth System Model (CESM) has been showed to be sensitive to the size of ice particles in Arctic LLMPCs; owing to a stronger cloud-phase feedback the larger the ice particles (Tan and Storelvmo, 2019). Achieving a complete understanding of ice microphysical processes in Arctic LLMPCs is therefore necessary in order to reach a correct representation of these unique clouds, together with their radiative effect, in climate models.

Millimeter-wavelength radars can effectively provide robust observational fingerprints to constrain cloud microphysical processes (e.g., Kollias et al., 2007; von Terzi et al., 2022). Cloud radars with Doppler capabilities have been widely used to gain insights into macrophysical characteristics (Shupe et al., 2006; Nomokonova et al., 2019), precipitation characteristics (Zhao and Garrett, 2008; Schoger et al., 2021), phase partitioning (De Boer et al., 2011; Kalesse et al., 2016; Griesche et al., 2020; Gierens, 2021), and dynamics and turbulence (Shupe et al., 2008; Mages et al., 2023) in Arctic clouds. The addition of polarimetric and multi-frequency capabilities can further expand the spectrum of obtainable fingerprints. Linear depolarization and dual polarization observations can in fact provide strong constrains for the shape and concentration of ice particles (Oue et al., 2015; von Terzi et al., 2022). At the same time, millimeter-radar observations at multiple wavelengths provide robust constraints for the size of ice particles: the ratio of the radar reflectivities measured at two separate wavelengths, named the dual-wavelength ratio (DWR), can be related to the characteristic size of the ice particle population (Hogan et al., 2000; Dias Neto et al., 2019). Furthermore, ice microphysical processes are highly sensitive to temperature (Pruppacher and Klett, 2012), hence matching radar observations with accurate thermodynamic information can further constrain the microphysical processes generating the radar fingerprints (Barrett et al., 2019; von Terzi et al., 2022).

Here, we present a quality-controlled dataset including dual-wavelength, polarimetric, Doppler cloud radar observations of LLMPC events taken at the high Arctic site of Ny-Ålesund, Svalbard, Norway, from 10 October 2021 until 31 December 2022. Observations from a zenith-pointing 94 GHz cloud radar, and a 35 GHz dual-polarization scanning cloud radar are complemented with thermodynamic retrievals from a microwave radiometer, cloud base height from a ceilometer and wind fields from large-eddy simulations. The objective is to provide a quality-controlled, ready-to-use, comprehensive dataset for microphysical studies of LLMPCs, taken at a site where such observations were not available until now. To our knowledge,





similar datasets featuring multi-frequency polarimetric Doppler cloud radar observations in the Arctic, coupled with thermo-dynamic information, are only available at the site of Utqiaġvik (Verlinde et al., 2016), in the American high Arctic, and for the MOSAiC expedition (Shupe et al., 2022), which took place in the central Arctic. We thus believe that this dataset will be a valuable addition to the already publicly available datasets, providing a tool for microphysical studies of Arctic LLMPCs in the European high Arctic, where Arctic Amplification has been shown to be the most intense (Dahlke and Maturilli, 2017). The dataset was published in Chellini et al. (2023) and is freely available at www.doi.org/10.5281/zenodo.7803064.

## 2  Measurement site and instruments

Observations were carried out at the observatory of the Arctic research station AWIPEV in Ny-Ålesund (Fig. 1), located at 79°N along the coast of the Kongsfjorden, a fjord on the west side of Spitzbergen, the main island of the Svalbard archipelago. The area features a mountainous terrain, with peaks reaching 700 m. The observatory is located at 11 meters above sea level, within 500 meters from the sea and 2 km from the Zeppelin mountain (556 meters high). The Kongsfjorden is surrounded by several glaciers, and the surface is of the tundra type. During the observational period the sea surface inside the fjord remained for the most part ice-free.

The orography often channels surface wind along the fjord axis, at 120° (southeasterly) (Beine et al., 2001; Esau and Repina, 2012; Graßl et al., 2022) . The surface wind layer thickness has been estimated to be comparable to the height of the surrounding mountains, with a yearly cycle of 500 m in summer and 1000 m in winter (Esau and Repina, 2012). The free troposphere above generally displays a westerly flow (Maturilli and Kayser, 2017).

Mean monthly surface air temperature peaks in July, at 5.8°C, and has its minimum in March at -12.0°C (Maturilli et al., 2013). Median monthly values of integrated water vapor (IWV) display a similar yearly cycle, with a March minimum of 3 kg m$^{-2}$ and a July maximum of 13 kg m$^{-2}$ (Nomokonova et al., 2020). Temperature and moisture inversions are a frequent feature of the lower troposphere at Ny-Ålesund, observed respectively in 75 and 84% of the daily radiosondes launched between 1993 and 2014 (Maturilli and Kayser, 2017).

Cloud characteristics at the site have been reported by a number of studies, including Nomokonova et al. (2019), Vassel et al. (2019), Koike et al. (2019), Nomokonova et al. (2020), Gierens et al. (2020), Ebell et al. (2020) and Chellini et al. (2022). Nomokonova et al. (2019) reported a frequency of occurrence of 20.6% for single-layer MPCs, with no restriction on height, while Gierens et al. (2020) estimated the occurrence of single-layer LLMPCs lasting more than 1 hour to be 23%. Chellini et al. (2022) observed an occurrence of LLMPCs lasting more than 1 hour, with no restriction on the number of liquid layers, of 30.7%.

### 2.1  Cloud radars

Three cloud radar systems were used to collect the data included in the dataset. All systems are frequency-modulated continuous-wave (FMCW) radars, manufactured by Radiometer Physics GmbH (RPG): JOYRAD-94 and MiRAC-A, which are 94 GHz



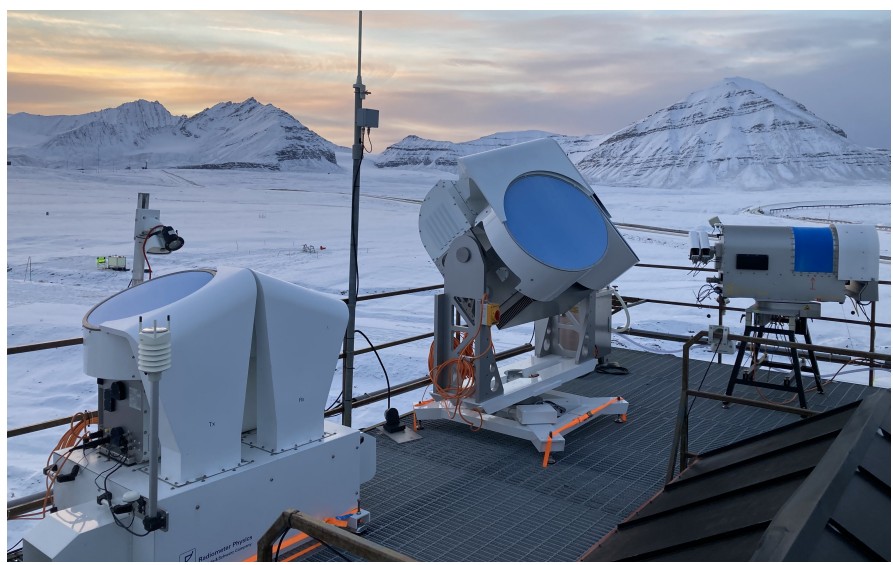

**Figure 1.** Measurement set-up at the AWIPEV observatory in Ny-Ålesund. From left to right: JOYRAD-94, NyRAD-35, HATPRO.

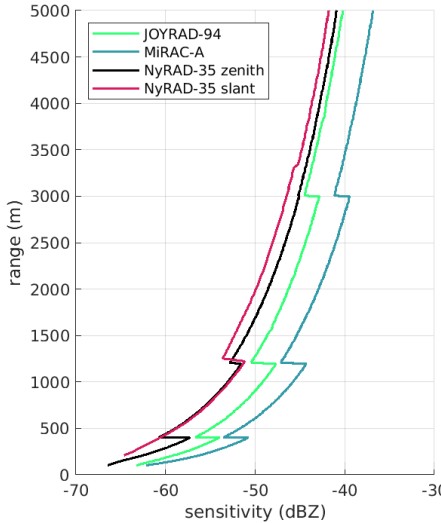

**Figure 2.** Median profiles of the sensitivity limit of all radar systems and chirp programs used in the study.



single-polarization zenith-pointing radars (model RPG-FMCW-94-SP, hereafter referred to as W-band), and NyRAD-35, a 35
90 GHz dual-polarization scanning radar (model RPG-FMCW-35-DP, hereafter referred to as Ka-band).

Contrary to the more commonly used pulsed radars, FMCW radars transmit a continuous wave, which is modulated in frequency around the central frequency (i.e. 35 or 94 GHz for the radars employed in this study). The signal is modulated in a saw-tooth pattern, with each individual ramp step typically referred to as chirp. Multiple chirps are combined into a chirp sequence, and target ranging is performed by determining the frequency difference between the transmitted and the received
chirp sequence, named intermediate frequency (IF). In practice, due to a limited IF bandwidth, a number of chirp sequences is required in order to sample the whole troposphere. The radar transmits the chirp sequences in succession, with the total time resolution being determined by the sum of the sampling time needed for each chirp sequence. We refer to a group of chirp sequences as chirp program, and the timestamp indicated in the files refers to the end of each execution of a chirp program. The exact chirp program settings can be defined by the user, and the values used to collect the data are reported in Table 2. The
sensitivity profiles associated with the different chirp programs and instruments are displayed in Fig. 2.

The Ka-band was operated throughout the whole dataset period, while JOYRAD-94 was operated until 22 June 2022, when it was replaced by MiRAC-A. The technical characteristics of JOYRAD-94 are described in Küchler et al. (2017), while MiRAC-A is described in Mech et al. (2019). The details regarding the processing of data from both W-band radars, in particular the noise removal and Doppler aliasing correction, are presented in Küchler et al. (2017). The two W-band systems are similar
in most aspects, the major differences being the larger beam width of MiRAC-A (0.85°) compared to JOYRAD-94 (0.5°), and the longer time resolution of JOYRAD-94, which despite using the same chirp program settings as MiRAC-A during the study period, needs an extra 0.4 seconds to reinitialize the measurements before the first chirp sequence of each chirp program. Nonetheless, the effective integration time is the same for both instruments in a given chirp sequence, and ranges between 0.27 and 0.37 s. Data from both W-band radars have already been used in a number of studies, including Dias Neto et al. (2019),
Wendisch et al. (2019), Gierens et al. (2020), and Schween et al. (2022).

Data from NyRAD-35, the Ka-band radar, are instead first used in this dataset, and more details on the instrument, as well as the data processing will be given. NyRAD-35 is a dual-polarization simultaneous transmission simultaneous reception (STSR) radar, with elevation-scanning capabilities. The scanner can perform full 180° scans in elevation, and is operated at a constant azimuth of approximately 27°, i.e. along the north-north-east to south-south-west direction. STSR radars receive at horizontal
and vertical polarization, but transmit a signal that is linearly polarized along the 45° direction between vertical and horizontal directions. This allows both for linear depolarization ratio (LDR) observations in zenith and typical dual-polarization variable observations at low elevations. The radar was set up to perform a scan cycle that includes zenith observations and lower elevation observations (30°-40° elevation). For optimal matching with the W-band radars, the chirp program during zenith observations was set up with the same range resolution as the W-band radars. Matching Ka- and W-band observations in zenith
allows for the calculation of the DWR, in order to obtain information on the characteristics size of the ice particle population. The slanted observations are instead used to obtain dual-polarization variables, such as differential reflectivity ($Z_{DR}$) or specific differential phase ($K_{DP}$). A different chirp program is used for low elevation observations, with a constant range resolution of 29.8 m. Similarly to the W-band radars, the noise level is determined with the widely-used approach by Hildebrand and Sekhon





|  | JOYRAD-94 | MiRAC-A | NyRAD-35 |
|---|---|---|---|
| Central frequency | 94.0 GHz | 94.0 GHz | 35.0 GHz |
| Time res. | 2.4 s | 2.0 s | 6.0/4.0 s |
| Beam width (half power) | 0.5° | 0.85° | 1.0° |
| Polarimetry | single pol. | single pol. | dual pol. (STSR) |
| Orientation | zenith only | zenith only | scanning |
| Data availability | 10.10.21-22.06.22 | 23.06.22-31.12.22 | 10.10.21-31.12.22 |
| Variables | Doppler moments | Doppler moments | Doppler moments, LDR (zenith), polarimetry (low elevation) |
| Aliasing corrected | yes | yes | no |

**Table 1.** Selected specifications of the three radar systems used in this study: two W-band cloud radars, JOYRAD-94 and MiRAC-A, and a Ka-band cloud radar, NyRAD-35.

(1974). Aliasing is instead not corrected, since the Nyquist range is large enough not to produce aliasing in zenith observations
in most conditions (see Table 2), while velocity information is not of interest at lower elevations, since a radial component of
the horizontal wind is present in the observed Doppler velocity. At low elevation Ka-band observations are affected by ground
clutter, namely artefacts caused by backscattering of the signal by the ground surface, and range folding, i.e. the incorrect
ranging of targets located beyond the maximum unambiguous range of the measurements. The correction of such artefacts is
described in detail in sections 4.3 and 4.4.

## 2.2 Microwave radiometer

Radar observations are complemented with thermodynamic information from a Humidity and Temperature Profiler (HATPRO;
generation 2) microwave radiometer (Rose et al., 2005). HATPRO records brightness temperatures (BTs) in 14 channels,
6 located in the K-band close to the 22-GHz water vapour absorption line, one located in the Ka-band in the atmospheric
window at 31.4 GHz, and the remaining 7 in the V-band close to the 60-GHz oxygen absorption line. Liquid water path
(LWP), integrated water vapour (IWV) and temperature profiles are retrieved from BT observations using the multivariable
linear regression approach described in Nomokonova et al. (2019). Temperature in particular is retrieved using the approach by
Crewell and Löhnert (2007), which exploits BTs observed at multiple elevations, so called boundary-layer scans, to improve
the accuracy of the retrieval, especially in the lowest 1 km. The accuracy of this technique was assessed by Gierens et al.
(2020) in single-layer LLMPCs at Ny-Ålesund against radiosondes, and they reported an RMSE of 0.7 K at the surface, that
increases to 1.6 K (2.0 K) at the base of the liquid layer (cloud top) of the LLMPC. HATPRO performs a 30-minute scan
cycle, starting every hour at :00 and :30 past; it is composed of a full 360° azimuth scan at 30° elevation lasting 2-3 minutes,
followed by a boundary layer scan for the temperature retrieval, and zenith observations for the remaining part of the scan
cycle (ca. 23 minutes). During the zenith observations a second boundary layer scan is performed, approximately 15 minutes
after the previous one. The LWP and IWV values included in the dataset are recorded during the azimuth scans and the zenith



| | W-band | Ka-band zenith | Ka-band off-zenith |
|---|---|---|---|
| **Chirp Seq. 1:** | | | |
| Min range [m] | 100 | 100 | 200 |
| Max range [m] | 400 | 400 | 1243 |
| Range res. [m] | 3.2 | 3.2 | 29.8 |
| Nyquist range [m s$^{-1}$] | 5.1 | 23.3 | 28.4 |
| Doppler res. [m s$^{-1}$] | 0.020 | 0.045 | 0.055 |
| Integration time [s] | 0.37 | 1.47 | 0.37 |
| Tot. sampling time [s] | 0.64 | 1.51 | 1.24 |
| **Chirp Seq. 2:** | | | |
| Min range [m] | 400 | 400 | 1243 |
| Max range [m] | 1200 | 1200 | 3329 |
| Range res. [m] | 7.5 | 7.5 | 29.8 |
| Nyquist range [m s$^{-1}$] | 5.1 | 22.0 | 19.1 |
| Doppler res. [m s$^{-1}$] | 0.020 | 0.043 | 0.075 |
| Integration time [s] | 0.27 | 1.47 | 0.78 |
| Tot. sampling time [s] | 0.48 | 1.60 | 0.98 |
| **Chirp Seq. 3:** | | | |
| Min range [m] | 1200 | 1200 | 3329 |
| Max range [m] | 3000 | 3000 | 6309 |
| Range res. [m] | 9.7 | 9.7 | 29.8 |
| Nyquist range [m s$^{-1}$] | 3.2 | 10.8 | 9.7 |
| Doppler res. [m s$^{-1}$] | 0.013 | 0.042 | 0.076 |
| Integration time [s] | 0.37 | 1.74 | 0.69 |
| Tot. sampling time [s] | 0.50 | 1.94 | 0.85 |
| **Chirp Seq. 4:** | | | |
| Min range [m] | 3000 | 3000 | 6309 |
| Max range [m] | 13000 | 13000 | 14000 |
| Range res. [m] | 23.8 | 23.8 | 29.8 |
| Nyquist range [m s$^{-1}$] | 3.2 | 8.1 | 4.7 |
| Doppler res. [m s$^{-1}$] | 0.025 | 0.063 | 0.037 |
| Integration time [s] | 0.27 | 0.64 | 0.73 |
| Tot. sampling time [s] | 0.38 | 0.95 | 0.92 |

**Table 2.** Chirp program settings for the radar observations reported in this study: the chirp program reported for W-band was used both for JOYRAD-94 and MiRAC-A. Two separate chirp programs were used for NyRAD-35 (Ka-band) depending on elevation.



observations: off-zenith values are multiplied by the sine of the elevation angle to obtain the corresponding vertical value. The temperature profiles included in the dataset are only taken from the boundary layer scans, and interpolated to a finer time resolution.

Retrievals from HATPRO are quality controlled via two separate approaches: by eliminating data points were rain was recorded by the instrument's weather station, and by using a spectral consistency check. The first approach consists in elim-

150 inating all data points when the weather station detected precipitation and temperature above 0°C, as rain depositing on the instrument's radome might invalidate the BT observations. A further sanity check is performed by retrieving the BT for each channel individually based on BTs observed by all other channels: if the simulated and observed BTs do not match within a certain tolerance, the data point is assumed invalid and removed form the dataset. The exact criteria used to remove unreliable data points were determined empirically, and are as follows:

– at K- and Ka-band the sum of the absolute differences between channels 2 through 7 is larger than 3 K,

– at V-band the sum of the absolute differences between all channels is larger than 10 K,

– at V-band 4 or more channels have absolute differences larger than 2 K.

Data are removed if any of the listed criteria is satisfied. This approach is especially useful after periods of rain, when precipitation has stopped, but the radome is still wet, thus effectively rendering the BTs unreliable.

## 2.3 Cloudnet target classification product

In order to obtain cloud phase information, used to determine whether a certain cloud event is a LLMPC, and information on the height of the liquid base of LLMPCs we use the Cloudnet target classification product (Hogan and O'Connor, 2004; Illingworth et al., 2007). The product combines data from the W-band cloud radars, microwave radiometer, and a ceilometer (model Vaisala CL51; Maturilli and Ebell (2018)), together with output from the ICOsahedral Nonhydrostatic weather model

(ICON; Zängl et al. (2015)), in its global numerical weather prediction mode (ICON-NWP), to classify the phase associated with radar and ceilometer echoes. It distinguishes between: cloud droplets, supercooled cloud droplets, and cloud ice, as well as drizzle or rain. The algorithm determines the presence of liquid at sub-zero temperatures based on ceilometer echoes, as layers composed of liquid droplets produce intense backscattering of the ceilometer signal. We employ the phase information in the Cloudnet product to determine the presence of LLMPCs, as described in section 4.2. Additionally, we extract from the

Cloudnet product the height of the base of the lowest liquid layer in LLMPCs, and include it in the dataset.

## 2.4 ICON-LEM setup

Additional meteorological variables not obtainable via our observations were needed in the dataset, especially humidity and pressure profiles, necessary to correct the radar data for attenuation. For these purposes we extract background thermodynamic profiles, as well as wind fields, from the output of the ICON model, ran in its large-eddy version (ICON-LEM; Dipankar et al.,

2015; Schemann et al., 2020). ICON-LEM uses a 3D Smagorinsky turbulence scheme, and is ran in a circular domain with





110 km diameter centered around Ny-Ålesund, 600 m horizontal resolution and 100 vertical levels. The domain of the LEM is nested in a larger domain, where ICON is run in the numerical weather prediction (ICON-NWP) mode, which is used as forcing. The full details on the model set-up can be found in Schemann and Ebell (2020) and Kiszler et al. (2023). Compared to their set-up a slight modification in the microphysical scheme was applied, which does not significantly impact the variables

used in this dataset. Kiszler et al. (2023) in particular validated the model output against radiosondes and HATPRO retrievals and found strikingly good agreement between simulated and observed wind and humidity fields. Wind fields from ICON-LEM are interpolated to the same range and time resolution as the radar data, and included in the dataset. The wind data from the first three hours after the start of each simulation (daily at midnight) are not included, as they might not be reliable.

## 3 Derivation of polarimetric variables from STSR-mode cloud radars

STSR cloud radars are still not largely used, and the derivation of certain polarimetric variables, especially the correlation coefficient between the received signals at horizontal and vertical polarization $\rho_{\mathrm{HV}}$ and the linear depolarization ratio (LDR), can be approached with a variety of methods. Hence, in this section we provide a brief summary of how we derive typical polarimetric variables from STSR radar observations. Cloud radars operating at STSR mode (also called hybrid mode) measure the so-called coherency matrix $\mathbf{B}$. The calculation of the coherency matrix follows Eq. 3.146 in Bringi and Chandrasekar

(2001), given for the processing of a pulse train in weather radars. In contrast, cloud radars compute the coherency matrix for each spectral line of a Doppler spectrum. For a spectral line at a given range the coherency matrix $\mathbf{B}$ can be expressed as function of the received electric field:

$$\mathbf{B} = \langle \mathbf{E}\mathbf{E}^{\dagger} \rangle = \begin{pmatrix} \langle E_h E_h^* \rangle & \langle E_h E_v^* \rangle \\ \langle E_v E_h^* \rangle & \langle E_v E_v^* \rangle \end{pmatrix}, \tag{1}$$

where $\mathbf{E}$ is a column vector that indicates the complex amplitude of the received electric field, and $E_h$ and $E_v$ respectively

its horizontal and vertical components. The complex conjugate is indicated with $^*$, the conjugate transpose with $\dagger$, and the $\langle \cdot \rangle$ brackets indicate averaging across multiple samples. The elements of the spectral coherency matrix can then be expressed as:

$$\mathbf{B} = \begin{pmatrix} B_{hh} & \dot{B}_{hv} \\ \dot{B}_{hv}^* & B_{vv} \end{pmatrix}, \tag{2}$$

where $B_{hh}$ and $B_{vv}$ are real numbers proportional to the power received at horizontal and vertical polarization, respectively; $\dot{B}_{hv}$ is instead the complex covariance between the two received components. Here all components of $\mathbf{B}$ are rescaled to have

the typical units of the equivalent radar reflectivity factor in linear scale, i.e. $\mathrm{mm}^6 \ \mathrm{m}^{-3}$.

Before radar variables are calculated, spectral lines containing signal have to be detected. For the detection we use the approach described in (Görsdorf et al., 2015, see Sec. 3.3. therein). For the threshold we used 6 standard deviations of noise above the mean spectral noise level. In this dataset the mean spectral noise level determination is performed with the algorithm by Hildebrand and Sekhon (1974). In the following $\widetilde{B}_{hh}$ and $\widetilde{B}_{vv}$ indicate a spectral line exceeding the threshold and having the

mean spectral noise level subtracted in the horizontal and vertical channel, respectively. Reflectivity at horizontal and vertical



polarization directions can be expressed as:

$$Z_{eH} = \sum \widetilde{B}_{hh}, \; Z_{eV} = \sum \widetilde{B}_{vv}, \tag{3}$$

where $\sum$ indicates summation over all spectral lines detected in a Doppler spectrum. Differential reflectivity can be expressed as:

$$Z_{DR} = \frac{\sum \widetilde{B}_{hh}}{\sum \widetilde{B}_{vv}} = \frac{Z_{eH}}{Z_{eV}}. \tag{4}$$

Following von Terzi et al. (2022) we include in the dataset the maximum spectral $Z_{DR}$ as well:

$$sZ_{DR\,max} = \max(sZ_{DR}) = \max\left(\frac{\widetilde{B}_{hh}}{\widetilde{B}_{vv}}\right), \tag{5}$$

where the maximum is calculated across all Doppler spectral lines with $\widetilde{B}_{hh,vv} > 0$. Similarly to von Terzi et al. (2022), in order to reduce the noise-induced variance of the variable we calculate the maximum in eq. 5 only across spectral lines with spectral 215 signal-to-noise ratio (sSNR) higher than 10 dB in both polarimetric channels. Here sSNR is defined as the ratio of $\widetilde{B}_{hh,vv}$ over the corresponding mean spectral noise level, i.e. the total noise power divided by the number of Doppler bins. Furthermore, in order to achieve higher consistency between different chirp sequences $sZ_{DR} = \widetilde{B}_{hh}/\widetilde{B}_{vv}$ is linearly interpolated (in log-scale) on a common Doppler resolution of 0.1 m s$^{-1}$ before calculating the maximum.

The phase shift between the horizontal and vertical components of the received electric fields can be expressed as:

$$\Phi_{DP} = \text{phase}\left(\sum \dot{B}_{hv}^{*}\right) \doteq \arctan\left(\frac{\text{Im}(\sum \dot{B}_{hv}^{*})}{\text{Re}(\sum \dot{B}_{hv}^{*})}\right), \tag{6}$$

where $\text{Re}(z)$ and $\text{Im}(z)$ indicate the real and imaginary parts of the complex number $z$. Note that here the summation is performed only over spectral lines where the signal is detected at both polarizations. By calculating the half range derivative of $\Phi_{DP}$ one obtains the specific differential phase $K_{DP}$. $K_{DP}$ typically displays large noise-induced fluctuations (e.g. Trömel et al., 2013), hence additional processing needs to be performed in order to reduce its variance. Here we calculate $\Phi_{DP}$ by 225 including a sSNR threshold of 10 dB in the summation in eq. 6, and $K_{DP}$ is further smoothed by applying a moving average in range (10 range gates, or 298 m) and time (5 minutes), similarly to von Terzi et al. (2022).

### 3.1 Correlation coefficient calculation

The correlation coefficient between the horizontal and vertical components of the received electric field $\rho_{HV}$ is typically computed with the formula (Bringi and Chandrasekar, 2001, eq. 6.110a):

$$\rho_{HV} = \frac{|\sum \dot{B}_{hv}|}{\sqrt{(\sum B_{hh}) \cdot (\sum B_{vv})}}. \tag{7}$$

Note that here the summation is performed only over spectral lines where the signal is detected at both polarizations. According to Bringi and Chandrasekar (2001, Eq. 6.122 therein) $\rho_{HV}$ calculated this way is prone to a bias due to the signal-to-noise ratio



(SNR). This often leads to apparent signatures in $\rho_{\mathrm{HV}}$ which are not caused by cloud microphysics but rather by a low SNR. One possible solution to this problem is to subtract the corresponding mean noise level from the power estimates in the denominator of Eq. 7:

$$\rho_{\mathrm{HV}} = \frac{|\sum \dot{B}_{hv}|}{\sqrt{(\sum \widetilde{B}_{hh}) \cdot (\sum \widetilde{B}_{vv})}}. \tag{8}$$

Myagkov and Ori (2022) noted however that this approach often leads to inaccurate $\rho_{\mathrm{HV}}$ values, due to the removed noise level being an estimate which might not exactly correspond with the true noise power. In some occasions $\rho_{\mathrm{HV}}$ values computed with eq. 8 might in fact exceed 1. Here we propose a modification to eq. 8 that has the potential to circumvent the effects of an incorrect noise level estimation on $\rho_{\mathrm{HV}}$. We use eq. 7 instead of eq. 8, and only perform the summation on spectral lines where $\rho_{\mathrm{HV}}$ is not strongly affected by noise contributions. We approach this by searching for spectral lines where the contributions from non-coherent antenna coupling (Myagkov et al., 2015) as well as particle depolarization exceed contributions from noise. We decompose the coherency matrix $\mathbf{B}$ into non-coherent and fully coherent components following Born and Wolf (1975, Sec. 10.8 therein):

$$\mathbf{B} = 0.5A\mathbf{I} + \mathbf{C}, \text{ with } \det(\mathbf{C}) = 0, \tag{9}$$

where $A$ is the non-coherent power, $\mathbf{I}$ is a $2 \times 2$ unity matrix, $\mathbf{C}$ is the coherency matrix of the fully coherent component, and det is the matrix determinant.

Following Myagkov and Ori (2022, Sec. 3.1 therein) we represent the measurements in the basis in which the coherency matrix is diagonal, i.e. the orthogonal linear components of the received signal are not correlated:

$$\mathbf{D} = \begin{pmatrix} D_{cc} & 0 \\ 0 & D_{xx} \end{pmatrix} = 0.5A\mathbf{I} + \mathrm{tr}(\mathbf{C}) \begin{pmatrix} 1 & 0 \\ 0 & 0 \end{pmatrix}, \tag{10}$$

where tr is the matrix trace. The elements $D_{cc}$ and $D_{xx}$ can be computed as the eigenvalues of the coherency matrix $\mathbf{B}$. As can be seen from Eq. 10, the element $D_{xx}$ contains only the non-coherent component. The non-coherent component in general includes contributions by noise, by depolarization by cloud particles, and by the presence of non-coherent antenna coupling. By determining the spectral lines with signal in $D_{xx}$ (same procedure as for $B_{hh}$ and $B_{vv}$), we identify spectral lines with considerable contribution from non-coherent antenna coupling and depolarization by cloud particles. The correlation coefficient is then calculated as in Eq. 7 with the summations performed only over spectral lines where $D_{xx}$ exceeds the threshold over the noise level.

We calculate the linear depolarization ratio (LDR) in zenith from $\rho_{\mathrm{HV}}$ following the approach given in Galletti and Zrnic (2011) and Galletti et al. (2011). Assuming the reflection symmetry (Nghiem et al., 1992) and that $Z_{\mathrm{DR}}$ is equal to 1 (in linear units), which is typically the case at vertical elevation, LDR can be computed by combining Eq. 12 in Galletti and Zrnic (2011) with Eq. 10 in Galletti et al. (2011), to obtain:

$$\mathrm{LDR} = \frac{1 - \rho_{\mathrm{HV}}}{1 + \rho_{\mathrm{HV}}}. \tag{11}$$

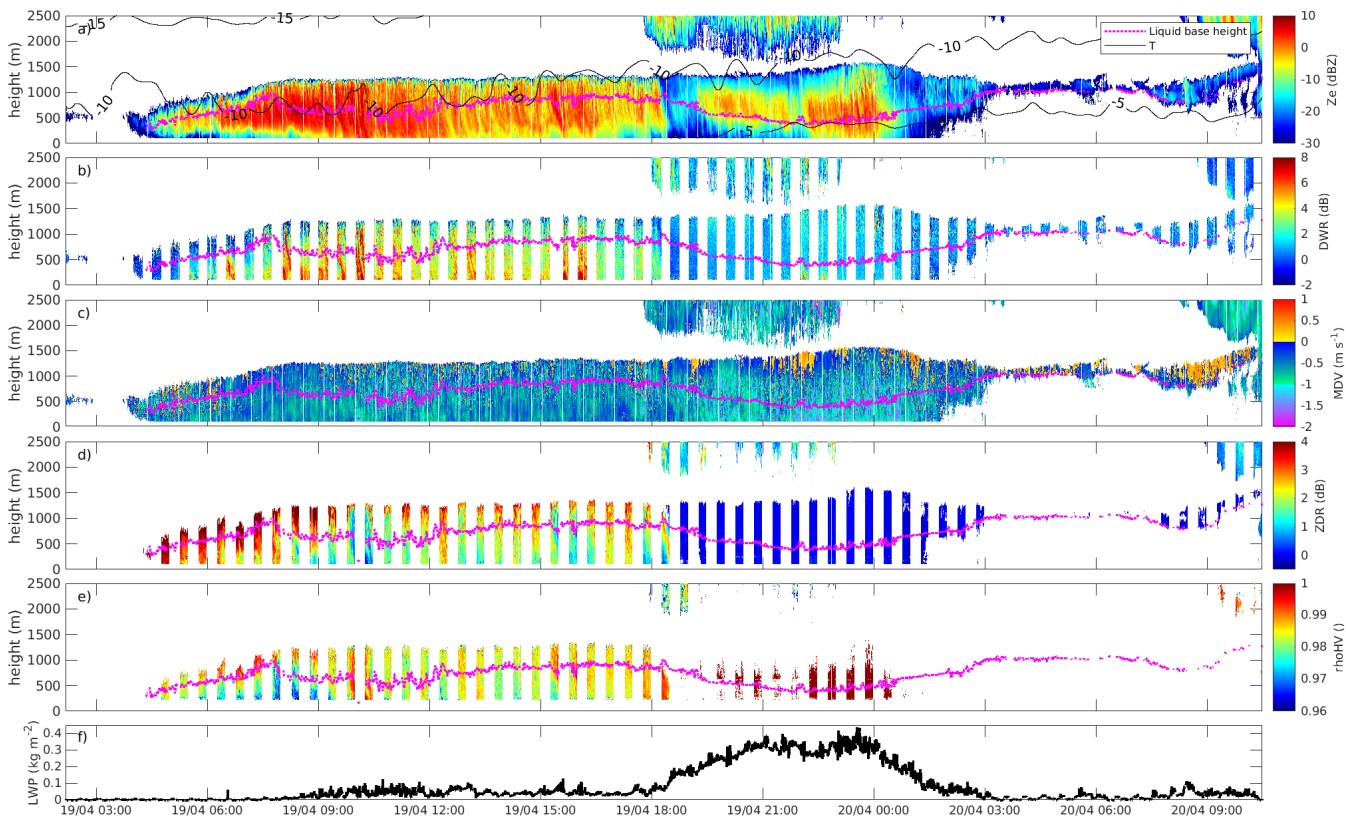

**Figure 3.** LLMPC event detected on 19-20 April 2022. Panels respectively display: reflectivity from W-band radar with temperature contours overlayed (a), dual-wavelength ratio (b), mean Doppler velocity from W-band radar (c), differential reflectivity (d), correlation coefficient (e), liquid water path (f). All radar variables were recorded at zenith elevation, except for differential reflectivity and correlation coefficient that were recorded at 30° elevation. The dotted magenta line on panels a through e indicates the base height of the lowest liquid layer.

We found that Eq. 11 at times produces LDR values below the integrated cross-polarization ratio (ICPR) of the instrument, which is the minimum LDR value that the radar can measure (Chandrasekar and Keeler, 1993; Myagkov et al., 2015). The

radar manufacturer in fact declares that the ICPR is between -30 and -35 dB, and eq. 11 at times produces values below -35 dB, which is likely attributable to the uncertainty in the approximation given by Eq. 11. This uncertainty is caused by the signal variability due to non-coherent scattering. This effect is however is not an issue, since it is widely accepted in the cloud and weather radar community that high quality LDR observations are obtained when the ICPR is close to or lower than -30 dB. Therefore we set all LDR values below -30 dB obtained from eq. 11 to -30 dB.



## 4 Data processing

### 4.1 Dataset structure

The dataset is structured in the following way: each individual file contains data from all sources for one or multiple LLMPC event; multiple LLMPC events are included in the same file if they are less than 4 hours apart. The criteria used to detect LLMPC events are described in the next section. The variables include Doppler moments from W-band and Ka-band in zenith, LDR in zenith from Ka-band, polarimetric variables from Ka-band at low elevation (30-40°), corrections applied to the radar data (calibration, liquid and gas attenuation), LWP, IWV and temperature profiles retrieved from HATPRO, liquid base height from the ceilometer, and horizontal wind information from ICON-LEM. All variables except low-elevation observations of the Ka-band radar are brought to the same time and range resolution of Ka-band zenith observations, for easier matching of data from different instruments. All corrections have already been applied to the data, and are included in case the user is interested in reconstructing the original uncorrected data. An example of an LLMPC event is given in Fig. 3, which displays a subset of the variables included in the dataset; the signatures displayed in the case study are further commented in section 5.

### 4.2 LLMPC event detection

LLMPC events were identified following the approach by Chellini et al. (2022), which is here summarized. A set of criteria are applied to the data to determine whether a given cloud event is a LLMPC:

1. Cloud top is at or below 2500 m. If multiple cloud layers are detected, at least one needs to have its top at or below 2500 m.

2. Cloudnet indicates presence of both liquid and ice in the cloud layer(s) with top below 2500 m.

3. Liquid and ice are detected by Cloudnet for at least 60 minutes with gaps allowed. The total duration of gaps in either ice or liquid phase needs to be lower than one sixth of the total duration of the LLMPC event.

During intense precipitation events ceilometer data is not always available. This is due to snow accumulating on the ceilometer aperture and leading to complete attenuation of the signal. Under these conditions Cloudnet only detects ice clouds, and criterion 2 is not satisfied even though a LLMPC might be present. These conditions are detected via the quality flags associated with ceilometer observations in the Cloudnet product. Under these circumstances LWP retrievals from HATPRO are instead used to determine the presence of the liquid phase, requiring that LWP is larger than 10 g m$^{-2}$.

All data corresponding to a given LLMPC event are collected in the same file, together with the previous and following two hours, in order to provide information on the conditions leading to the onset and cessation of events. Two events are included in the same file if the two hours following a given event overlap with the two hours preceding the next event.

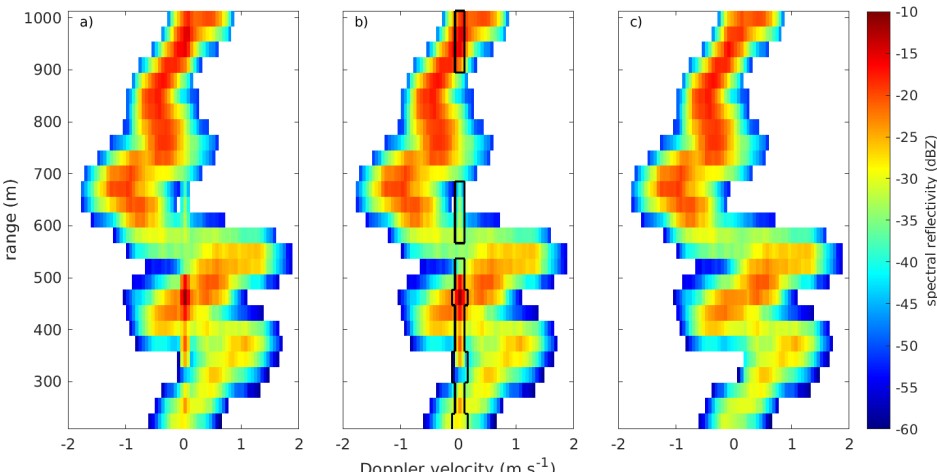

**Figure 4.** Example of the ground clutter mitigation procedure applied to Doppler spectra recorded at 30° elevation on 15 October 2022 at 21:23:38 UTC. Panel a depicts the original Doppler spectrum, while in panel b the contour indicates the Doppler bins detected as affected by ground clutter. Panel c displays the final spectrum after clutter removal.

## 4.3   Ka-band: ground clutter mitigation

We found the low elevation observations of the Ka-band radar to be contaminated with ground clutter, namely the presence of artefacts caused by backscattering of the radar signal by the ground. We correct the data using an approach similar to that developed by Williams et al. (2018). Thanks to the lack of moving clutter-generating targets, all clutter signal is only found either in the $0 \text{ m s}^{-1}$ Doppler velocity spectral line, or in nearby lines. An example of a Doppler spectrum affected by ground clutter is shown in Fig. 4 panel a, while panel c displays the same spectrum after clutter removal. Indicating the spectral reflectivity in dBZ of the spectral line at $0 \text{ m s}^{-1}$ as $\text{sZ}_{\text{e}}(i_0)$, the presence of clutter is determined when the two following criteria are satisfied:

1. $\text{sZ}_{\text{e}}(i_0) - \frac{1}{4}[\text{sZ}_{\text{e}}(i_0 - 2) + \text{sZ}_{\text{e}}(i_0 - 1) + \text{sZ}_{\text{e}}(i_0 + 1) + \text{sZ}_{\text{e}}(i_0 + 2)] > 2$ dB;

2. $\text{sZ}_{\text{e}}(i_0) < -15$ dBZ.

The second criterion was determined empirically, since we never observed ground clutter characterized by a spectral reflectivity higher than -15 dBZ. If ground clutter is identified, the three spectral lines $i_0 - 1$, $i_0$, and $i_0 + 1$ are removed, and replaced with linearly interpolated values between $\text{sZ}_{\text{e}}(i_0 - 2)$ and $\text{sZ}_{\text{e}}(i_0 + 2)$. This procedure is applied independently to $\widetilde{B}_{hh}$, $\widetilde{B}_{vv}$, and the modulus of the off-diagonal component $|\dot{B}_{hv}|$. If clutter is identified in $|\dot{B}_{hv}|$, the corresponding spectral lines in phase($\dot{B}_{hv}$) are also removed, and replaced with the median value of phase($\dot{B}_{hv}$) across all spectral lines without clutter, and above the noise level. In the lowest 20 range gates (i.e. from 200 to 796 m range) we found, at times, the criteria listed above not to be sufficient, because in some cases ground clutter might also be present in the spectral lines $i_0 - 2$ and $i_0 + 2$, and might display

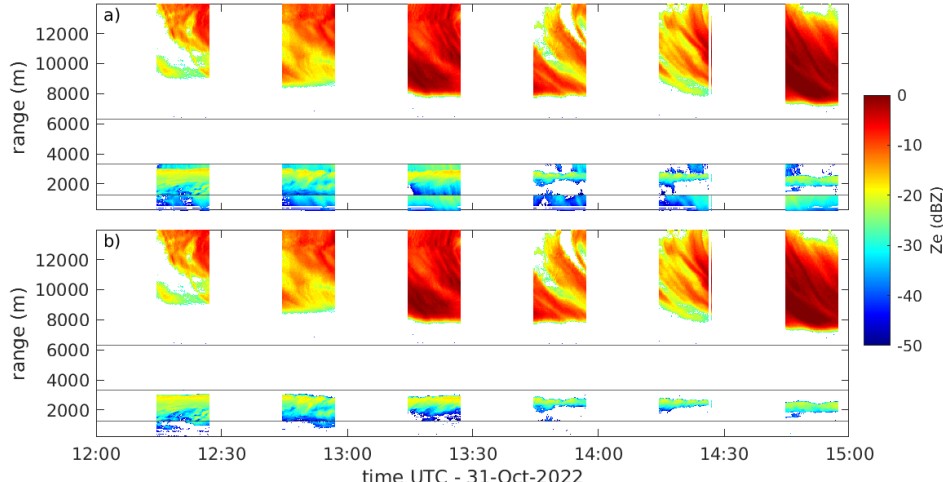

**Figure 5.** Reflectivity time-height display recorded at 30° elevation on 31 October 2022. Horizontal lines indicate boundaries between chirp sequences. Panel a depicts the original reflectivity values, affected by range folding (in the two lowest chirp sequences), while panel b displays reflectivity after the removal of range folding.

spectral reflectivity in the 0 m s$^{-1}$ spectral line higher than -15 dBZ. After applying the criteria listed above, a second set of criteria is applied in the lowest 20 range gates, to detect possible residual clutter:

1. $\mathrm{sZ_e}(i_0) - \frac{1}{4}[\mathrm{sZ_e}(i_0 - 3) + \mathrm{sZ_e}(i_0 - 2) + \mathrm{sZ_e}(i_0 + 2) + \mathrm{sZ_e}(i_0 + 3)] > 2$ dB.

2. $\mathrm{sZ_e}(i_0) < 5$ dBZ.

For range gates satisfying these criteria the interpolation is also extended to $i_0 - 3$, $i_0 + 3$.

## 4.4   Ka-band: range folding correction

At low elevation angles, range folding was sometimes observed in the Ka-band data in the two lowest chirp sequences. Range folding is the incorrect ranging of targets located at ranges larger than the maximum unambiguous range of the radar measurements. The maximum unambiguous range for the two lowest chirp sequences is in fact 10000 m, which at 30° elevation corresponds to a height of 5000 m, above which it is likely to observe clouds. In RPG FMCW radars the apparent range $r$ of a 325 target affected by range folding is:

1. $r = R_{unamb} - (R - R_{max})$, when the actual range $R$ is $R_{unamb} < R < R_{unamb} + R_{max} - R_{min}$,

2. $r = R - (R_{unamb} + R_{max})$, when the actual range $R$ is $R_{unamb} + R_{max} + R_{min} < R < R_{unamb} + 2 \cdot R_{max}$,

where $R_{max}$ and $R_{min}$ are the maximum and minimum ranges of the chirp sequence, as indicated in table 2, and $R_{unamb}$ is the maximum unambiguous range. We found that in the first folding scenario phase($\dot{B}_{hv}$) assumes unrealistic values, and 330 events displaying folding of this type are easily corrected by including a sanity check $|\mathrm{phase}(\dot{B}_{hv})| < 50°$. Folding scenarios of

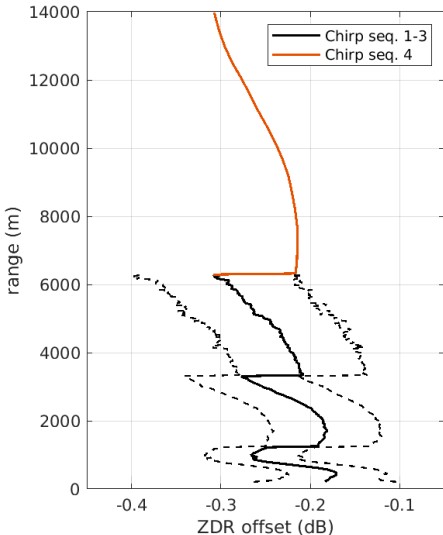

**Figure 6.** Differential reflectivity ($Z_{DR}$) offset profile. Black lines indicate values obtained for chirp sequences number 1 to 3, while the orange curve was extrapolated for chirp sequence 4. Dashed lines indicate the median value plus and minus one standard deviation.

the second type are more challenging to correct. We only observed folding of this type to occur in the first chirp sequence, and echoes affected by range folding in the first sequence are captured without range folding in the fourth chirp sequence. Therefore we developed the following procedure to remove range folding from the first chirp sequence when echoes are present in the fourth sequence:

1. Spectra from the first chirp sequences are interpolated to the same Doppler resolution as the fourth chirp sequence.

2. The interpolated spectra are then artificially brought to the same Nyquist range as the fourth chirp sequence by artificially velocity-folding them, as the Nyquist velocity of the fourth chirp sequence is lower than that of the first chirp sequence.

3. Spectral reflectivity from the fourth chirp sequence is rescaled by $r^2 \cdot R^{-2}$ to match the reflectivity of the range folded signal.

4. Spectra from the two chirp sequences are subtracted.

5. The resulting spectra are interpolated back to the original Doppler resolution.

6. Possible residual values below sensitivity and isolated spectral lines are removed.

An example of an event with folding is shown in Fig. 5, before and after the correction.

### 4.5 Ka-band: $Z_{DR}$ and $\Phi_{DP}$ calibration

Radar systems receiving at two polarization directions might display slight offsets in $Z_{DR}$ and $\Phi_{DP}$, due e.g. to differences in antenna gain along the two polarimetric directions, or in the waveguides and feedhorns of the two polarimetric channels. In





FMCW systems such offsets might further display a dependency on range, as they might depend on the IF, as well as range resolution. Here we evaluate such offsets by exploiting $Z_{DR}$ and $\Phi_{DP}$ observations in zenith, as we can expect them to be close to 0 dB and 0° respectively. Since such offsets can be dependent on the chirp program used, we evaluate them using the same chirp program that is used for low-elevation observations. When switching elevation positions the Ka-band radar in fact performs RHI scans (not included in this dataset), that use the same chirp program. In order to derive the offsets, we compile $Z_{DR}$ and $\Phi_{DP}$ profiles recorded at elevations between 85 and 95°, and calculate median profiles. The $Z_{DR}$ offset obtained with this approach is displayed in Fig. 6 in black: the values range between -0.31 and -0.17 dB, with a standard deviation of 0.06 dB, that is constant with range. Due to the low number of observations in zenith at range above 6 km, the offset profile for chirp sequence number 4 could not be obtained. We instead used the $Z_{DR}$ offset profile from chirp sequence number 1, expressed it as function of IF, and then mapped it to chirp sequence number 4 by matching the IF to the range gates of chirp sequence number 4. The result of this procedure is shown in red in Fig. 6. The complete offset profile was then subtracted from all $Z_{DR}$ and $sZ_{DR\,max}$ data.

Contrary to $Z_{DR}$, the offset for $\Phi_{DP}$ was found not to vary with range, except for the lowest 8 range gates, and to vary with season, reaching values close to -4.0° in winter, and close to -1.5° in the summer. The latter dependency is likely attributable to thermal expansion and contraction of components of the radar hardware. Due to the variability of the offset we decide not to correct for it, as the absolute value of $\Phi_{DP}$ is anyhow not of interest, but rather its change with range, expressed by $K_{DP}$. Nevertheless, we choose to remove $\rho_{HV}$ and $\Phi_{DP}$ (and hence $K_{DP}$) data from the 8 lowest range gates, as it might contain spurious signals.

## 4.6  Ka- and W-band: temporal matching

In order to obtain reliable DWRs, accurate matching and calibration of the Ka- and W-band data needs to be performed. The chirp programs of the W-band and the zenith observations of the Ka-band were set up to have the same vertical grid. Temporal matching was performed by bringing the higher temporally-resolved W-band data to the same temporal resolution as the Ka-band observations. The central time of each sample was calculated for each Ka- and W-band chirp sequence, and each Ka-band sample was matched with the nearest available W-band sample, up to a maximum time difference equal to the time resolution of the W-band measurements. By performing this temporal matching we found indications of inconsistencies between the timestamps of the two radars. We attributed them to slightly incorrect timestamps of the Ka-band radar due to errors in the recording of the timestamps by the instrument software. Daily time offsets were determined by shifting the Ka-band data in time, and selecting the time shift that minimized the variance in DWR. The attempted time shifts ranged from -120 seconds to +60 seconds, in steps of 0.25 seconds.

## 4.7  Ka- and W-band data: calibration

Biases in observed reflectivity might arise due to changes in the calibration constant of the instruments. Hence we evaluated the calibration of all radar systems. While the Ka-band radar was recently calibrated by the manufacturer in August 2021 using the technique from Myagkov et al. (2020, Sec. 4 therein), the W-band radars have not undergone any calibration in the recent



380 years. We evaluated the calibration of the Ka-band radar with a widely adopted disdrometer-based approach (e.g., Williams et al., 2005; Myagkov et al., 2020), while the W-band radars were cross-calibrated against the Ka-band radar using a DWR-based approach (Dias Neto et al., 2019; Chellini et al., 2022). Reflectivity from Ka-band zenith observations were compared with forward-simulated reflectivities obtained from drop size distributions (DSDs) measured during rain events by a Parsivel disdrometer (Löffler-Mang and Joss, 2000), which is located on the same platform as the radars, approximately 6 meters away

from them. The rain events used for calibration were selected in July, August and September 2022 based on the following criteria:

- Surface temperature (from nearby weather station) is higher than 5°C, and Parsivel detects liquid precipitation.

- In order for the disdrometer measurements to be representative of the drop population, rain rate $\geq 0.1$ mm h$^{-1}$ (following Williams et al. (2005)), and measurements contain at least 25 samples per minute.

- DSDs contain particles larger than 1 mm. Such criterion was included so that possible evaporation of the drops between the chosen radar range gate and the disdrometer doesn't affect the forward simulated reflectivities (following Myagkov et al. (2020)).

- Rain events satisfying the previous criteria need to last at least 1 hour, with gaps allowed for a total of one sixth of the duration of the event.

Reflectivities were forward simulated from the DSDs with the T-matrix method (Waterman, 1965; Leinonen, 2014), using a drop shape model from Thurai et al. (2007), with Gaussian distributed canting angles (with 0° mean value and a 10° standard deviation), following Huang et al. (2008). Attenuation due to liquid was simulated as well, and was subtracted from the forward simulated reflectivity values. The forward-simulated reflectivities were compared with observed reflectivities from the range gates between 120 and 150 m. The comparison was performed by calculating the median observed and simulated reflectivity

across all calibration events. The calibration offset for the Ka-band radar was found to be -0.14 dB; since this value is well within the uncertainty associated with reflectivity measurements, we do not apply any corrections to Ka-band reflectivity, and consider it already well-calibrated.

The W-band radars are instead cross-calibrated using the Ka-band as reference. The approach consists in exploiting observations of low-reflectivity ice clouds or light snowfall, to ensure the presence of mostly Rayleigh scatterers, which produce

similar reflectivities at Ka- and W-band (e.g., Dias Neto et al., 2019; Tridon et al., 2020). The data used in the cross-calibration were selected using the following criteria:

- Cloudnet indicates the presence of ice only clouds.

- LWP retrievals from HATPRO are lower than 10 g m$^{-2}$.

- Reflectivity from the Ka-band is between -30 dBZ and -10 dBZ, to ensure the presence of mostly Rayleigh scatterers
(following Dias Neto et al. (2019)).




| Date | 12-Oct-2021 | 28-Oct-2021 | 14-Nov-2021 | 11-Dec-2021 | 07-Jan-2022 | 03-Feb-2022 |
|---|---|---|---|---|---|---|
| Offset [dB] | $0.7 \pm 1.1$ | $1.3 \pm 1.2$ | $0.5 \pm 1.1$ | $0.5 \pm 1.0$ | $0.4 \pm 1.1$ | $0.5 \pm 1.1$ |
| Date | 02-Mar-2022 | 29-Mar-2022 | 26-Apr-2022 | 20-May-2022 | 23-Jun-2022 | 21-Jul-2022 |
| Offset [dB] | $0.6 \pm 1.3$ | $0.5 \pm 1.2$ | $0.2 \pm 1.0$ | $0.7 \pm 1.1$ | $3.0 \pm 1.1$ | $3.3 \pm 1.1$ |
| Date | 19-Aug-2022 | 15-Sep-2022 | 12-Oct-2022 | 10-Nov-2022 | 05-Dec-2022 | 01-Jan-2023 |
| Offset [dB] | $2.7 \pm 1.0$ | $2.9 \pm 1.1$ | $2.4 \pm 1.0$ | $2.7 \pm 1.1$ | $1.9 \pm 0.9$ | - |

**Table 3.** Calibration offsets, and associated uncertainties, obtained for the W-band cloud radars. Positive values indicate an underestimation of reflectivity by the radar. Each offset was computed for the period starting on the date indicated, and ending on the day before the date indicated for the following offset. Values before 22 June 2022 refer to JOYRAD-94, while the remaining values refer to MiRAC-A.

DWR distributions for data points satisfying these criteria are compiled for a number of periods lasting approximately one month, and the mode of each distribution is taken as W-band reflectivity offset for the corresponding period. All offsets, together with the associated uncertainties, are indicated in Table 3. The uncertainty is calculated by taking the left-side standard deviation of the distribution with respect to the mode, as the right side might be affected by non-Rayleigh effects.

## 4.8 Ka- and W-band: gas and liquid attenuation correction

Millimeter-wavelength radar signal undergoes non-negligible attenuation due to atmospheric gases, especially molecular oxygen and water vapour, and to hydrometeors. Typical path-integrated attenuation for ice and snow at W-band has been estimated as 0.9 dB km$^{-1}$ for ice water path (IWP) values of 1 kg m$^{-2}$ (Nemarich et al., 1988): due to the typically far lower values of IWP observed in LLMPCs at Ny-Ålesund, in 75% of cases below 60 g m$^{-2}$ (Gierens et al., 2020), we deem attenuation due to frozen hydrometeors negligible and do not correct for it. On the other hand attenuation due to liquid hydrometeors is highly relevant at millimeter wavelengths, especially at W-band (e.g., Tridon et al., 2020). Hence we estimate the liquid water content (LWC) and correct the data for liquid attenuation.

Here we correct for gas attenuation using thermodynamic profiles from the ICON-LEM simulations, and simulating the associated attenuation with the Passive and Active Microwave Transfer model (PAMTRA; Mech et al., 2020). PAMTRA computes 2-way path-integrated attenuation profiles due to molecular nitrogen, molecular oxygen, and water vapour, taking temperature, humidity and pressure profiles as input.

Attenuation due to liquid hydrometeors is estimated using a combination of observational and model data. We derive a theoretical scaled adiabatic LWC profile based on liquid base height and cloud top height from Cloudnet, LWP from the microwave radiometer and thermodynamic profiles from the ICON-LEM output. LWC profiles close to scaled adiabatic have been in fact reported in LLMPCs across the Greenland and Norwegian seas by Mioche et al. (2017). The procedure consists in calculating an adiabatic LWC profile starting at the liquid base, until cloud top, using the thermodynamic profiles from ICON-LEM. The obtained adiabatic LWC profile is then rescaled by a vertically-constant factor to match the retrieved LWP. Liquid attenuation is then simulated using the water dielectric constant model by Rosenkranz (2014) and assuming that liquid droplets absorb in the Rayleigh regime, using eq. 5.18 in Bohren and Huffman (2008). This approach relies on the assumption



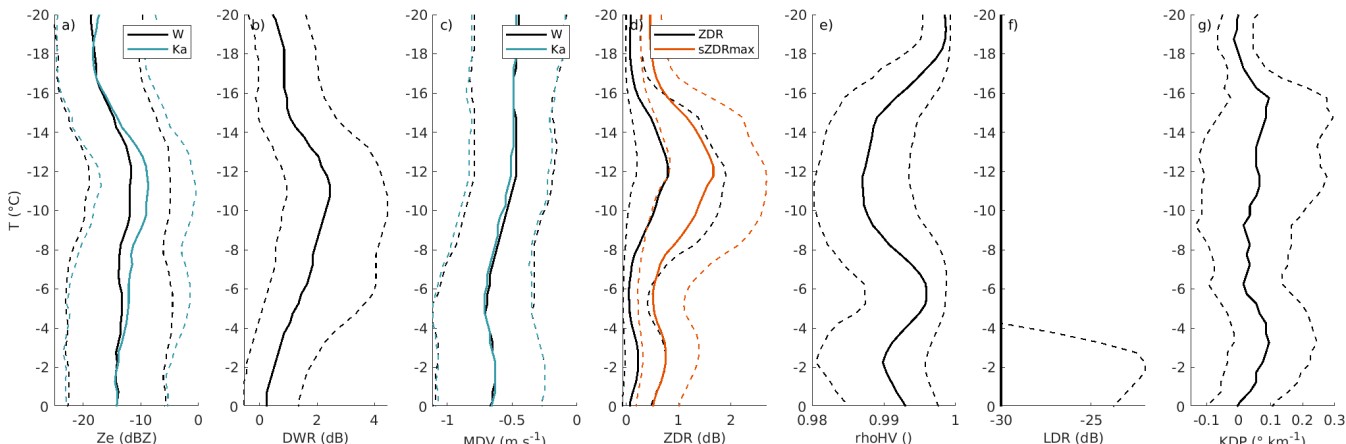

**Figure 7.** Contoured Frequency by Temperature Diagrams of radar variables in low-level MPCs detected at Ny-Ålesund during the study period. Solid lines indicate the median value in each temperature bin, while dashed lines indicate the 0.25 and 0.75 quantiles. The variables included are: radar reflectivity at W- and Ka-band (a), dual-wavelength ratio (b), mean Doppler velocity at W- and Ka-band (c), differential reflectivity and maximum spectral differential reflectivity (d), correlation coefficient (e), linear depolarization ratio (f), and specific differential phase (g). Temperature is obtained from HATPRO retrievals, temperature bins are 0.5°C wide.

that one continuous liquid layer is present between the liquid base indicated by the ceilometer and cloud top. This might not always be the case. Nonetheless, we deem this approach to be most sound possible with the information that we have available, since the ceilometer only provides the base height of the lowest liquid layer. Additionally the correction is set to 0 dB if LWP is below 20 g m$^{-2}$. It needs to be highlighted that no liquid correction is applied if either the liquid base information or LWP information are not available. Flags that indicate whether the liquid attenuation correction is available and has been applied

have been included in the files. We recommend that the user makes full use of these flags to exclude from quantitative analyses reflectivity values that have not been corrected for liquid attenuation. This is especially relevant when calculating the DWR.

## 5  Characterization of ice particle formation and growth in LLMPCs at Ny-Ålesund

In this section we present and discuss the case study shown in Fig. 3, as well as a statistical analysis of the temperature dependence of observational fingerprints of microphysical processes obtained from radar measurements. Fig. 3 displays a

number of variables recorded during a LLMPC event observed on 19-20 April 2022. The panels of the Figure display radar reflectivity from the W-band radar (a), dual wavelength ratio (b) calculated by subtracting the W-band reflectivity from the Ka-band reflectivity (both in log-scale), mean Doppler velocity from the W-band radar (c), differential reflectivity (d) and correlation coefficient (e) from 30°-elevation Ka-band observations, and LWP (f) retrieved from HATPRO observations. Liquid base height from the ceilometer is overlaid on all panels, while temperature contours are overlaid on panel a. From the onset

of the event until 17:30 on 19 April $Z_{DR}$ displays high values, above 2 dB, indicative of the presence of oblate ice particles. The temperature of the liquid layer, slightly colder than -10°C, is compatible with the growth of plate-like particles (e.g.,

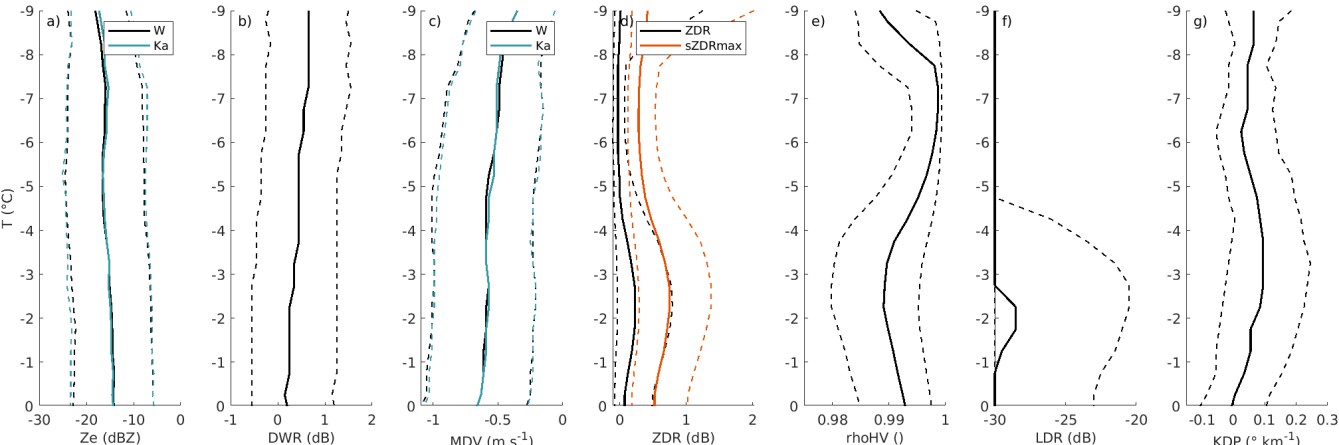

**Figure 8.** Same as in Fig. 7, but limited to only LLMPC events with cloud-top temperature warmer than -10°C.

Bailey and Hallett, 2009). Ice particles in fact nucleate in the liquid layer, hence their habit is strongly dependent on liquid layer temperature (Myagkov et al., 2016; Chellini et al., 2022). $Z_{DR}$ decreases as particles sediment, and DWR increases, indicating the onset of aggregation, which increases the size of the ice particles (which in turn enhances DWR), and renders

the particles more spherical (which in turn lowers $Z_{DR}$). In this first period MDV displays values close to 1 m s$^{-1}$, which are compatible with the presence of aggregates of plate-like particles (Locatelli and Hobbs, 1974; Heymsfield and Westbrook, 2010; Karrer et al., 2020). As already reported by Chellini et al. (2022) aggregation already onsets within the liquid layer of the LLMPC. After 17:30 the characteristics of the LLMCP change dramatically: LWP increases from values lower than 100 g m$^{-2}$ to values close to 400 g m$^{-2}$, and precipitation is mostly composed of small, fast-falling, symmetrical particles. This is

indicated respectively by MDV reaching values close to 2 m s$^{-1}$, $\rho_{HV}$ close to 1, and $Z_{DR}$ close to 0 dB. These factors together suggest the presence of rimed particles and/or drizzle. Interestingly this sudden change in precipitation regime is accompanied by cloud-top temperatures (CTT) rising to values slightly warmer than -10°C, and a higher cloud with base at 1500 to 2000 m. Temperatures between -10 and -8°C in particular have been associated with the growth of isometric particles with higher riming efficiency (Fukuta and Takahashi, 1999; Myagkov et al., 2016), that might explain the dramatic change in radar signatures at

17:30. After 23:00 on April 19th, LWP decreases again and the LLMPC reduces to a thin layer a few hundred meters deep (after 01:00 on April 20th), which does not produce any precipitation, as indicated by the liquid base from the ceilometer being close to the lowest range gates where radar reflectivity is displayed. The layer is likely composed mostly by liquid droplets, as reflectivity is below -20 dBZ.

## 5.1 Statistical analysis: temperature dependence of radar signatures

Ice microphysics is greatly dependent on temperature (Pruppacher and Klett, 2012), as such we can expect that observational fingerprints of microphysical processes display characteristic temperature dependencies. In order to illustrate these dependencies, in Fig. 7 we show contoured-frequency by temperature diagrams (CFTD; Yuter and Houze Jr, 1995) of most radar



variables introduced in the previous sections. Such diagrams illustrate how distributions of observed variables vary as temperature changes. In the Figure, radar variables for the whole study period are matched with temperature from HATPRO retrievals,

and median and quantiles of each variable are then calculated in 0.5°C-wide temperature bins. In order to avoid contamination by other cloud types, only timestamps flagged as containing a LLMPC, and only range gates below the cloud top of the LLMPC are included. The latter selection is performed by determining the lowest radar range gate without signal above the liquid base in each column; all range gates above are ignored, so that higher clouds do not contribute to the statistics. Only data points where the liquid attenuation correction is available are included in the reflectivity and DWR CFTDs.

As already noted by Myagkov et al. (2016) and Chellini et al. (2022), and highlighted when commenting the case study in Fig. 3, in LLMPCs the habit of the ice particles produced is fully determined by the temperature of the liquid layer, where the ice particles nucleate and grow by depositional growth. This is in contrast to deeper cloud systems, where ice habits are less predictable due to particles growing in a number of temperature regimes as they sediment. The first temperature regime that we would like to highlight in Fig. 7 is the so-called dendritic-growth zone, namely the region between -20 and -10°C

where plate-like particles grow (e.g., Bailey and Hallett, 2009; Bechini et al., 2013; von Terzi et al., 2022): inside this regime stellar or fern-like dendritic particles are observed under liquid saturation conditions, and especially between -16 and -12°C (Takahashi, 2014). Rapid growth and aggregation of dendritic particles in this temperature regime has been widely reported (Takahashi, 2014; Barrett et al., 2019; Dias Neto et al., 2019). Dendritic particles in fact aggregate efficiently due mechanical entanglement of their dendritic branches (Connolly et al., 2012). Fingerprints of dendritic growth and subsequent aggregation

can be especially inferred by examining the dual-wavelength, Doppler, and polarimetric radar variables in Fig. 7: DWR displays a steady increase starting at -15°C and peaking at -11°C, while MDV slightly increases in the same temperature region, displaying median values close to -0.6 m s$^{-1}$; $Z_{DR}$ and $sZ_{DR\,max}$ both start increasing and reach their maxima at a slightly colder temperature than DWR (-16 and -12°C respectively), while $K_{DP}$, especially its 0.75 quantile, starts increasing at colder temperatures (-18°C), reaches its maximum at -15°C, then steadily decreases as particles fall towards warmer temperatures. In

agreement with Chellini et al. (2022), the observed DWR and MDV values are compatible with low-density dendrite aggregates (Locatelli and Hobbs, 1974; Heymsfield and Westbrook, 2010; Kneifel et al., 2015). Bands of enhanced $K_{DP}$ and $Z_{DR}$ at dendritic-growth temperatures have been widely reported in mid-latitude deep precipitating systems (Bechini et al., 2013; Schrom et al., 2015; Griffin et al., 2018; von Terzi et al., 2022). The maximum in $K_{DP}$ at -15°C can be in fact related to the increase in number concentration of small asymmetric ice particles, and as such, a $K_{DP}$ increase has been suggested to be a

prerequisite for the onset of aggregation (Moisseev et al., 2015). The enhancement of $Z_{DR}$ is instead typically interpreted as originating from rapid growth of plate-like particles (Schrom and Kumjian, 2016; Griffin et al., 2018; von Terzi et al., 2022). Interestingly, in typical mid-latitude systems the maximum in $Z_{DR}$ is observed above the maxima of $K_{DP}$ and DWR, and $K_{DP}$ is found to increase steadily from the top of the dendritic-growth zone until its base (von Terzi et al., 2022), while Fig. 7 paints a different picture. In the Figure the peak in $K_{DP}$, and therefore ice number concentration, at -15°C, is followed by increases

in DWR and $Z_{DR}$ at slightly warmer temperatures. These discrepancies might be connected to the limited depths associated with Arctic LLMPCs, compared to mid-latitude deep precipitating systems. Moreover, fragmentation of dendritic and stellar crystals has been widely suggested in literature (e.g., Rangno and Hobbs, 2001; Schwarzenboeck et al., 2009; von Terzi et al.,



2022; Pasquier et al., 2022), and it might be associated with the increase in ice number concentration suggested by the sharp enhancement in $K_{DP}$ at -15°C.

At CTT warmer than -10°C, LLMPCs at Ny-Ålesund typically produce smaller particles, characterized by DWR close to 0 dB (Chellini et al., 2022). This is confirmed by Fig. 8, which displays CFTDs of the same radar variables included in Fig. 7, but restricted to events with CTT warmer than -10°C. The Figure displays median DWRs lower than 1 dB, compatible with particles with sizes close to or smaller than 1 mm (e.g., Ori et al., 2020), fall velocities that steadily increase from -0.4 m s$^{-1}$ at -10°C to -0.7 m s$^{-1}$ at 0°C, typical for small ice crystals with low degree of riming. Prolate particles, such as columns and needles,

typically grow in this temperature regime (e.g., Bailey and Hallett, 2009; Myagkov et al., 2016), and can be easily detected via their enhanced LDR (Oue et al., 2015; Bühl et al., 2016; Li et al., 2021): panel f in Fig. 8 displays that LDR values higher than -20 dB occurred less than 25% of the cases at temperatures between -4 and 0°C, with frequency dramatically decreasing at colder temperatures. They do not seem to produce large aggregates, as DWR remains low. The increase in $K_{DP}$ between -6 and 0°C is compatible with the hypothesis of formation of needle particles in a portion of the cases, as it signals an increase in

number concentration of asymmetric particles. Interestingly we can draw a parallel between the increase in $Z_{DR}$ at dendritic growth temperatures, which is preceded by an increase in $K_{DP}$, and the increase in LDR at column growth temperatures, also preceded by an increase in $K_{DP}$. This might be due to the time needed for small $K_{DP}$-producing ice particles to grow either into larger $Z_{DR}$-producing plates or larger LDR-producing columns. At the same time, this second $K_{DP}$ enhancement region could be an indication of secondary ice production, as it has been reported in Arctic LLMPCs at these temperatures (e.g., Luke

et al., 2021; Pasquier et al., 2022). This second $K_{DP}$ enhancement zone has in fact been observed in mid-latitude systems only in a limited fraction of cases by von Terzi et al. (2022), and to a far lesser extent, which supports the hypothesis that the observed enhancement might be attributable to mixed-phase-related secondary ice production, such as droplet shattering or rime splintering. The high fall velocities associated with the higher quartile in MDV could be indicative of riming (e.g., Kneifel and Moisseev, 2020), which is also compatible with low $Z_{DR}$, as well as high $\rho_{HV}$, especially observed between -8

and -5°C.

## 6   Conclusions

We present a comprehensive long-term dataset of remote sensing observations of low-level mixed-phase clouds (LLMPCs), taken at the high Arctic site of Ny-Ålesund, Svalbard, Norway. The dataset in particular features dual-wavelength and polarimetric Doppler cloud radars observations, which are especially suited for ice microphysical studies. Cloud radar observations are complemented by thermodynamic retrievals from a microwave radiometer (temperature, liquid water path, and integrated

water vapor), liquid base height from a ceilometer, and wind fields from large-eddy simulations. LLMPCs were detected using criteria based on the persistence of ice and liquid phases, and only periods when LLMPCs were detected were included in the dataset. All variables have undergone extensive quality control, especially the cloud radar observations, that were calibrated, as well as corrected for attenuation, ground clutter, and range folding. Unreliable data from microwave radiometer retrievals





was also removed using precipitation information and a spectral consistency approach. All variables are brought to the same time and range grids, for easier matching of data originating from different instruments.

The large number of radar variables, coupled with thermodynamic, liquid base height and wind field data, included in the dataset allows for a comprehensive characterization of microphysical, as well as macrophysical, properties of LLMPCs. Microphysical studies will especially benefit from the combination of this dataset with data from the Video In-Situ Snowfall Sensor
(VISSS; Maahn et al., 2023), which was operated at Ny-Ålesund during the same period. Additionally, the large number of events included in the dataset (more than 600) enables to compile robust statistics, especially for studies of ice microphysical processes. We demonstrate this by performing a brief statistical analysis of the temperature dependence of Doppler, dual-wavelength and polarimetric radar variables. Expanding upon results from previous studies, we show that LLMPCs at temperatures between -18 and -10°C display dual-wavelength and polarimetric signatures compatible with the rapid increase
in number concentration, growth and subsequent aggregation of plate-like particles. We further show fingerprints suggesting the formation of precipitating prolate particles at temperatures warmer than -5°C. The analysis highlighted that LLMPCs are especially suited for process studies, as the ice habits involved can be easily inferred based on temperature, due to the limited depth of such clouds. This makes the dataset an ideal testing environment for lagrangian particle models and microphysical schemes.

## 7 Data availability

The full dataset has been published in Chellini et al. (2023), and is available at: www.doi.org/10.5281/zenodo.7803064.

*Author contributions.* GC processed the Ka-band radar data, including the developing of the range folding and ground clutter corrections, performed the calibration and attenuation correction of all radar systems, matched the data from all instruments and compiled the final files, all with the supervision of SK. RG processed the W-band data, and KE processed the HATPRO data and applied the Cloudnet target
classification. TK ran the ICON-LEM simulations, with the supervision of VS. GC, RG and PK operated the cloud radars, with support from AM. PK handled the data transfer and storage. AM provided guidance on radar operation and data processing. SK and GC designed the dataset, and GC wrote the manuscript and performed the analysis, with feedback from all authors.

*Competing interests.* The authors declare that they have no competing interests.

*Acknowledgements.* GC, RG, KE, TK, PK and VS gratefully acknowledge the funding by the Deutsche Forschungsgemeinschaft (DFG,
German Research Foundation)—Project no. 268020496—TRR 172, within the Transregional Collaborative Research Center "ArctiC Amplification: Climate Relevant Atmospheric and SurfaCe Processes, and Feedback Mechanisms (AC)[3]." Contributions by SK were partly funded



by the DFG under grant KN 1112/2-1 and KN 1112/2-2 as part of the Emmy-Noether Group "Optimal combination of Polarimetric and Triple Frequency radar techniques for Improving Microphysical process understanding of cold clouds" (OPTIMice).

The authors would like to express their gratitude to the Alfred Wegener Institute (AWI), and in particular Marion Maturilli and Christoph
Ritter, for sharing their HATPRO and ceilometer data. The authors further wish to thank the AWIPEV staff for assisting with the operation of the instruments. The authors wish to thank Bernhard Pospichal, for sharing and applying his spectral consistency method to evaluate the reliability of microwave radiometer retrievals, and Davide Ori, for producing the scattering tables used in the disdrometer-based calibration, and providing assistance when setting up the PAMTRA forward simulations. The authors would like to further acknowledge the discussions with Leonie von Terzi, which helped improve the processing of polarimetric radar data. GC and TK also acknowledge the support from the
Graduate School of Geosciences (GSGS) of the University of Cologne, as well as the Integrated Research Training Group (IRTG) of the $(AC)^3$ consortium.





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
