# Peer review of "Low-level mixed-phase clouds at the high Arctic site of Ny-Ålesund: A comprehensive long-term dataset of remote sensing observations"

_Earth System Science Data, 2023_

## Author Response (AR1)

We would like to thank the three anonymous referees for the time and effort they put in to review the first version of our manuscript. Their constructive comments enabled us to improve the quality and clarity of the manuscript. Please find our answers to the points raised below.

(Unless stated otherwise, all line numbers refer to the previously submitted version of the manuscript)

**Changes not requested by reviewers**

- *We found a mistake in the reported number of levels of the ICON-LEM simulations at line 176, which is 150 instead of 100. The mistake was corrected.*

- *The statement "...events included in the dataset (more than 600) …" at line 546 was inaccurate and has been corrected, it now reads: "...events included in the dataset (totalling 554) …".*

- *The sentence at lines 32-34 was removed since the referenced preprint seems to have been retracted.*

- *The references Kiszler et al. (2023) and Maahn et al. (2023) were updated.*

- *Further minor and purely stylistic changes, as highlighted in the track-changes document.*

**Reviewer 1**

- Section 2 The topographical situation around the observation location is described, but it would be better to give a figure showing the observation location in relation to the topography.

  *We agree with the reviewer and included a new figure depicting the local orography (Figure 2 in the newly submitted version).*

- Section 2.4 model setup In addition to the number of vertical layers, please information on vertical grid spacing and/or model top heigh. You describe "each simulation (daily at midnight)".  Does it mean that the simulation starts daily at midnight local time and integrate for 27 hours, and did you use simulation results between 3 and 27 hours? It would be better to describe it explicitly.

  *We agree with the reviewer that some information was missing or not clearly stated, and we added two sentences at lines 176-178:*
  *"The vertical grid spacing increases with height, starting at 20 m at the surface, increasing to 90 m at 2500 m height.  [...] Simulations start daily at 00 UTC, and run for 24 hours, until 00 UTC on the following day."*

- Section 4.2 As mentioned briefly in conclusions in section 6, please also summarize and describe basic quantities such as the number of cases, the distribution of durations and total hours extracted.

  *Following the reviewer's recommendation we added a new paragraph at line 295, it reads: "The total number of events detected amounts to 554, accounting for a total duration of 3426 hours. The events are seasonally distributed as follows: 135 events or 878 hours in*

*winter, 93 events or 670 hours in spring, 123 events or 662 hours in summer, and 224 events or 1371 hours in autumn, with seasons following the meteorological convention.”*

*A new sentence was also added at the end of the section, at line 310, it reads: “As a consequence, all 554 events are collected into 247 files.”*

**Reviewer 2**

- My main concern with this manuscript is that, apart section 5, is very oriented to a readership of radar experts. While this makes the paper complete and thorough from a technical viewpoint, I wonder what could be done to promote it to a larger community. My fear is that the geographical peculiarity of this dataset will not be exploited as the main users will be radar scientists and not polar-meteorology scientists. I am not suggesting to reduce the technical parts, but to reorganize the manuscript and put forward more physical interpretations (as done in Sec. 5) and to try to have always in mind a readership composed of weather radar data users and not necessarily weather radar data experts. As a simple example, a more detailed description in terms of microphysical interpretation of Figure 3 could provide direct meteorological inputs earlier in the manuscript.

  *We are very grateful to the reviewer for suggesting to promote the manuscript to a broader community. However, we would prefer not to re-organize the entire manuscript structure for the following reasons:*
  - *We believe that the main novelty of our dataset is the combination of dual-frequency and polarimetric radar observations, so far only available in the Arctic at the ARM site in Utqiaġvik and during the MOSAiC expedition. These observations allow for in-depth investigations of cloud and precipitation microphysical processes. In order to perform such investigations a high degree of expertise in radar observations is certainly needed and the many details included in the manuscript are in our opinion very important for a potential user to understand which corrections or post-processing has been applied.*
  - *A number of other datasets featuring cloud remote sensing observations collected at Ny-Ålesund are already available, which are more suited for the needs of other scientific communities outside the radar and cloud microphysics community. We would like to highlight especially the Cloudnet initiative (https://cloudnet.fmi.fi). Within Cloudnet W-band radar and microwave radiometer data, together with a number of retrievals and products (including the target classification product used in this dataset) have been published daily since 2016.*

- L11: Zenodo. While looking at the repository on Zenodo, I would recommend to add a little bit more information on the data.

  *We agree with the reviewer that the description of the dataset on Zenodo was very minimal, and it has now been expanded with practical information on the contents of the files.*

- L44: A recent paper on multi-frequency retrieval (with many references inside) is also: https://doi.org/10.5194/amt-16-911-2023 based on the ICE GENESIS field campaign (with some similarities with respect to your installation)

  *We thank the reviewer for suggesting the additional reference. The reference to the BAMS article introducing the ICE GENESIS campaign (https://doi.org/10.1175/BAMS-D-21-0184.1) was added at line 46.*

- L91: "more commonly used" I think it is not necessarily correct, it depends by the field of application. Cloud profilers are relatively commonly FMCW

  *We agree and removed the first part of the sentence: "Contrary to the more commonly used pulsed radars".*

- L101: I think an overview of the operation periods, with eventual gaps/issues, should be given at this moment (a table or some description)

  *We agree with the reviewer that the presence of data gaps should be indicated in the text. While the W-band radars and ceilometer were not affected by any major data gaps, a number of data gaps are present in Ka-band radar and microwave radiometer data. Technical issues affecting the quality of the data did not occur. We added a clarification at line 280: "The user should be aware that a number of data gaps are present in the Ka-band and microwave radiometer data. These are due to software and data transfer issues, which did not affect the quality of the collected data. Such data gaps are filled with NaN values in the files."*

- L112: I was used to STAR (Simultaneous Transmit And Receive), but STSR makes sense i guess

  *We agree that some readers might be familiar with other expressions, such as the STAR acronym or the "hybrid-mode radar" expression, and changed the text in parentheses at line 112, it now reads: "(STSR; also referred to as hybrid-mode, or STAR, simultaneous transmission and reception)". Following this change, the mention to the "hybrid-mode" expression at line 188 was removed.*

- L118: 30° and 40° would be considered as very large elevation angles for most of the common weather radars. Why did you not go down to 0° - 180°?

  *The decision to limit the observations to 30-40° was due to a number of reasons.When first installing the Ka-band radar we attempted lower elevations, but due to the presence of the surrounding orography (see newly added Fig. 2) observations at elevation lower than 30° were severely affected by ground clutter. Range folding would have also been more severe, and potentially impossible to mitigate. Additionally, the instrument measures spectrally-resolved polarimetric variables (see maximum spectral ZDR variable included in the files), and performing measurements at elevations close to 0° would not have allowed us to take advantage of this capability. Lastly, due to the high spatial variability that Arctic LLMPCs display (see e.g., Ruiz-Donoso et al. (2020; ACP)), reaching elevations close to 0° would have lead zenith and low-elevation measurements to observe very distant regions of the same LLMPC, with potentially vastly different characteristics. We determined 30° and 40° to be reasonable compromises, considering that other studies in the literature successfully used polarimetric variables observed at these elevations [e.g., Myagkov et al. (2016; AMT), Pfitzenmaier et al. (2018; ACP), von Terzi et al. (2022; ACP)]. We deem that no adjustments to the text are needed here.*

- L122: How is KDP estimated? Research showed that the estimation of KDP from PhiDP / PsiDP can be affected by large errors and biases, but these studies were conducted at X, C and S band. Is there anything published about KDP at Ka band ? What can be the effect of differential phase shift on backscattering (delta) at this frequency? Can you please comment about this aspect in the manuscript?

*The method used to estimate KDP is presented at lines 219-226 (226-238 in the newly submitted version). We are aware of the following study investigating KDP at Ka-band: Oue et al. (2017; JGR:A) - doi.org/10.1002/2017JD027717*

*Regarding the effect of differential phase shift at 35 GHz, we added the following sentences at line 226 when describing the approach used to estimate KDP: "Typically in weather radar processing the estimate of KDP has a higher degree of complexity than the method here adopted (e.g., Hubbert and Bringi, 1995). This is due to weather radar observations typically being characterized by much lower SNR compared to cloud radars, and to the need for the removal of the backscatter differential phase δ contribution to ΦDP (Hubbert and Bringi, 1995; Trömel et al., 2013). It has been shown that at cloud radar wavelengths δ produced by dry ice particles is negligible (Lu et al., 2015; von Terzi et al., 2022)".*

- L150: Are these conditions to be both satisfied? (there could still be rain at slightly negative temperatures)

  *Yes, both conditions are to be satisfied. We slightly changed the sentence to make this clearer. While we agree that these criteria might miss events characterized by liquid (or partly-liquid) precipitation, these are highly likely to be flagged by the spectral consistency approach.*

- L150: 0°C, is it intended as wet bulb temperature?

  *0°C is intended as dry-bulb temperature. While we understand that wet-bulb temperature is typically used for melting layer detection, the choice of using dry-bulb temperature is due to microwave radiometer observations being invalidated even by small amounts of liquid present on the instrument's radome. For this reason we preferred to opt for using dry-bulb opposed to wet-bulb temperature. We believe no further adjustments to the text are needed here.*

- Figure 3: Could you comment on the abrupt change in ZDR at all heights, visible around 18 UTC?

  *We believe that a discussion of the dramatic change in cloud characteristics at 18 UTC, including ZDR, is already present at lines 458-465 (lines 477-484 in the newly submitted version). We did not expand this discussion in the current iteration, though we are keen to do it if the reviewer indicates specific scientific questions they would like us to address.*

- L555: If possible to provide some codes to manipulate the data or to replicate some of the statistics shown in Sec. 5, I would recommend to do so.
  *We thank the reviewer for making such suggestion, and we are considering the preparation of a number of scripts that could be used to perform simple analyses based on the dataset. Unfortunately this is not possible during the current review iteration due to time constraints.*

**Reviewer 3**

- I think it would be helpful to include a few more sentences on the Cloudnet classification product (lines 160-170). While sources are cited for this method, it would be helpful if the authors briefly describe what the model is doing with the observations in order to discern the cloud phase, particularly since it is the LLMPC detection method. The text indicates that Cloudnet uses the ceilometer backscatter signal combined with the radar variables (lines

165-170). How does Cloudnet classification compare to other methods that do not use a model to identify MPCs?

*The determination of the liquid phase by Cloudnet is based on the attenuated backscatter coefficient profile β' observed by the ceilometer. It simply uses two thresholds, on β' itself, and on the gradient of β'. The paragraph was expanded to include this piece of information, and to briefly explain that this approach is very common in the literature.*

- In line 101, it is noted that the Ka-band was operating throughout the whole dataset period. This piece of information could be moved to the paragraph starting at line 111, where the Ka-band radar (NyRAD-35) is further discussed. As a reader, it would be clearer to see that note separate from the paragraph focused on the JOYRAD-94 and the MiRAC-A.

*We agree with the reviewer and moved the sentence to line 111. The first two sentences of the paragraph starting at lines 111 now read: "NyRAD-35, the Ka-band radar, was operated throughout the whole dataset period. Data from the instrument are used for the first time in this dataset, and more details on the instrument itself, as well as the data processing will be given."*